# Environmental Filtering and Dispersal Limitations Driving the Beta Diversity Patterns at Different Scales of Secondary Evergreen Broadleaved Forests in the Suburbs of Hangzhou

**DOI:** 10.3390/plants12173057

**Published:** 2023-08-25

**Authors:** Liangjin Yao, Bo Jiang, Jiejie Jiao, Chuping Wu

**Affiliations:** 1Zhejiang Academy of Forestry, Hangzhou 310023, China; lj890caf@163.com (L.Y.); jiangbof@126.com (B.J.); jjjjust@163.com (J.J.); 2Zhejiang Hangzhou Urban Forest Ecosystem Research Station, Hangzhou 310023, China

**Keywords:** the coexistence of species, beta diversity, diffusional limitation, species turnover, species richness differences

## Abstract

Subtropical suburban secondary evergreen broadleaved forests are essential in regulating the ecological environment’s quality and promoting urban sustainable development. In the suburbs of Hangzhou City, well-preserved secondary evergreen broadleaved forest communities were selected to establish a 6 ha forest dynamic monitoring plot. Community surveys and environmental factor measurements were conducted in this area. This study investigated the beta diversity patterns at different scales by considering the environmental and spatial factors to explore the driving beta diversity. Using a similar paired-site beta diversity decomposition method, the study aimed to investigate the differences in species composition and the mechanisms of multiple species coexistence within the secondary evergreen broadleaved forest communities. The results showed that the beta diversity of the suburban secondary evergreen broadleaved forest communities decreased with the increasing spatial scale. Both the dispersal limitation and the environmental filtering were found to drive the formation of beta diversity patterns in these subtropical suburban forests. At relatively smaller scales (<100 m), species turnover was found to determine the beta diversity patterns of the suburban secondary evergreen broadleaved forests. Dispersal limitation had a dominant influence at more minor scales, while the effect of environmental filtering gradually increased with scale, and the impact of the dispersal limitation decreased. The partitioning of the beta diversity in subtropical secondary evergreen broadleaved forests in China provides critical scientific insights into the spatial distribution patterns and changes in biodiversity. It offers valuable knowledge for the conservation and understanding of biodiversity maintenance in the region.

## 1. Introduction

Studies at the plant population level often focus on alpha diversity to assess the community diversity and explore intra-specific relationships, inter-specific relationships, and the spatial distribution patterns of populations [1,2]. In recent years, with the establishment of large-scale forest dynamic monitoring plots and increased attention to research scales [3], studies based on forest community beta diversity have gained increasing attention. Beta diversity refers to the degree of biodiversity and ecological functional differences among different habitat types in space. It helps understand ecological differences and interrelationships among forest communities, revealing forest ecosystems’ stability and environmental functions and investigating community assembly processes. Therefore, it has become one of the focal points in biodiversity research [4].

Different forest community compositions, environmental factors, and other factors affect beta diversity differently. Studies have found that forest communities with relatively affluent species composition exhibit more significant differences in species composition, resulting in higher beta diversity at the spatial scale [5]. Regions with higher soil organic matter content tend to have higher beta diversity [6]. Stevenson et al. [6] investigated the contributions of flooding and soil composition to beta diversity based on the species composition and distribution in lowland forest communities in northern South America. They found that changes in the flooding and soil nutrient content led to changes in the species composition, thereby influencing the beta diversity. He et al. [5] utilized data from the Global Tropical Forest Network and conducted a multivariate regression analysis to determine the impact of various environmental factors on the beta diversity. The study revealed that spatial variation (such as elevation, latitude, and distance) significantly influenced the beta diversity of tree species in tropical forests.

Meanwhile, the impact of environmental changes (such as climate, soil, and human disturbance) on the beta diversity was relatively minor. The magnitude of the beta diversity is closely related to the spatial scale. At more minor spatial scales, the beta diversity tends to be lower, while the beta diversity tends to be higher at larger spatial scales. Some researchers suggest that the scale dependence of beta diversity is likely caused by environmental heterogeneity and dispersal limitation [7]. Scudeller and Vegas-Vilarrubia [8] explored the distribution of tree species and beta diversity in the Igapo forest of the Brazilian Amazon. They found that the composition of the tree species and beta diversity in the Igapo forest was mainly influenced by the geographical distance and environmental factors rather than the altitude and flood history. They emphasized the need for careful consideration of the ecological and geographical factors and the spatial connectivity between different sites in the conservation and management of Igapo forests. Therefore, the essential factors influencing the beta diversity are the research scale, range, sampling interval, and objective conditions resulting from the latitude gradient, topography, and changes over time (such as climate change) that the community experiences [9,10].

The mechanisms underlying the formation and maintenance of beta diversity in communities may vary in different ecosystems and habitat conditions. At a local small scale, forest communities with similar habitat conditions undergo multilayer habitat filtering (e.g., available resources, spatial and environmental factors, etc.), resulting in ecological niches with similarities, and species within the community adopt convergent ecological strategies [11]. Therefore, it is believed that communities under similar habitat conditions have similar species compositions [3,12,13]. On the other hand, differences in environmental factors, resource utilization, and disturbance conditions between different habitats lead to variations in species distribution, thereby influencing the formation and maintenance of beta diversity in communities [14,15,16]. Interactions among species in different habitats also affect the beta diversity in communities. Species in forest communities that share similar resource and spatial requirements have similar ecological niches, resulting in niche overlap or redundancy within the community. This similarity-driven process leads species to evolve toward divergent ecological strategies, reducing competition intensity and facilitating the coexistence of multiple species [13,15,16,17].

Consequently, the competition, predation, and symbiotic relationships among species vary with habitat differences, leading to the diversity in species composition and maintaining differences between communities. Migration and species dispersal between different habitats also contribute to changes in species composition and influence the beta diversity in communities [16]. Historical factors also play a role in beta diversity. For instance, evolutionary history, past ecological events, and other factors between different habitats lead to differences in species composition and subsequently affect the formation and maintenance of beta diversity in communities [18,19]. The construction and maintenance of beta diversity are likely influenced by multiple processes, and competition and dispersal are believed to influence community assembly [12,20] jointly. The assembly patterns of different forest types at the regional scale represent a continuous dynamic process driven by ecological niche processes transitioning to neutral theory processes.

Suburban forests are essential components of urban ecosystems with diverse biodiversity. They play a crucial role in ecological regulation, environmental protection, and enhancing biodiversity, providing multiple ecological services, such as maintaining water circulation, soil formation, air purification, and climate regulation [21]. They form the foundation for modern urban ecosystems’ construction and sustainable development. Research on biodiversity in suburban forests helps understand species composition, population numbers, distribution ranges, habitat requirements, and ecological functions, providing a scientific basis for protecting and managing urban ecosystems. Located in the western outskirts of Hangzhou, Zhejiang Wuchaoshan National Forest Park is characterized by a complex topography and an extensive distribution of secondary evergreen broadleaf forests, making it an excellent study site to assess species diversity and investigate the functional and community maintenance mechanisms of suburban forests [22]. In this study, based on community and environmental data from long-term forest monitoring plots within Wuchaoshan National Forest Park, we researched the beta diversity patterns and driving forces of subtropical suburban secondary evergreen broadleaf forest communities at multiple scales. The study aims to address the following questions: (1) What is the relationship between beta diversity patterns of suburban secondary evergreen broadleaf forest plots and sampling scales? (2) What are the mechanisms underlying the formation of beta diversity in suburban secondary evergreen broadleaf forest communities? By deepening our understanding of species coexistence mechanisms in suburban forest communities, we can provide scientific evidence for protecting and improving the stability and ecological functions of suburban forest ecosystems.

## 2. Results

### 2.1. Community Beta Diversity and Its Components

The study found that beta diversity, its replacement components (Repl), and abundance difference (AbDiff) gradually declined with the increasing spatial scale in the Wuchaoshan site community. At the 10 m × 10 m scale, the beta diversity value was 0.651. As the study scale increased, the beta diversity value decreased to 0.408 at the 50 m × 50 m scale (Figure 1). Across the three scales examined in this study, the contribution of species replacement to beta diversity remained dominant. From small to large scales, the proportion of the gift of species replacement to beta diversity increased, accounting for 65.6%, 69.5%, and 78.7%, respectively. On the other hand, the contribution of the abundance difference decreased continuously, accounting for 34.4%, 30.5%, and 21.3%, respectively.

### 2.2. Variance Decomposition of Community Beta Diversity

The study found that as the scale of the investigation increased, the explanatory power of the pure environmental impact on beta diversity showed an increasing trend. In contrast, the explanatory power of the refined spatial structure showed a decreasing trend. The explanatory power of the ecological spatial systems showed a growing trend. The portion of beta diversity that remained unexplained (d) exhibited a decreasing trend (Table 1). Taking the 20 m × 20 m scale as an example, the combined explanatory power of the environmental spatial structure (b) and pure spatial structure (c) accounted for a significant portion of the beta diversity variation, with an explanatory power of 66.76%. On the other hand, the explanatory power of the pure environmental impact (a) was relatively low, accounting for only 1.31% of the beta diversity (Table 1, Figure 2).

## 3. Discussion

### 3.1. The Changing Pattern of Beta Diversity of a Suburban Secondary Evergreen Broadleaved Forest Community

The study revealed a decreasing trend in the beta diversity of the suburban secondary evergreen broadleaved forest community in the subtropical region as the scale of the investigation increased. This finding is consistent with earlier research on beta diversity in the forest, microbial, and animal communities, confirming the universality of the negative correlation between beta diversity and spatial scale [23,24,25]. A study by Legendre et al. [26] on subtropical forest communities in Gutianshan found that the beta diversity of forest communities decreased with increasing scale (grain size). The relative importance of environmentally driven filtering increased gradually after decomposition, while the relative importance of the dispersal limitation decreased, explaining the scale-dependent nature of the beta diversity. Environmental heterogeneity refers to the presence of different habitat types, habitat quality, and environmental factors at different spatial scales. These factors influence species’ survival and reproduction, leading to variations in species composition among other locations. At a local scale with small environmental heterogeneity, the differences in species composition between different sites are relatively small, resulting in a gradual change in beta diversity [27,28,29].

On the other hand, environmental heterogeneity is more remarkable at a large scale, leading to more pronounced differences in species composition among different forest communities and making the changes in beta diversity more complex. Dispersal limitation refers to the restriction of species migration due to geographical distance and habitat isolation, resulting in differences in species composition among different locations. At a small scale, dispersal limitation is relatively low, and the differences in species composition between other sites are relatively small, resulting in a gradual change in beta diversity [21,22,30]. However, dispersal limitation is more pronounced at a large scale leading to more apparent differences in species composition between different locations, resulting in a more distinct pattern of beta diversity [25,31].

In the study site’s suburban secondary evergreen broadleaved forest, species accumulation was rapid at small scales. However, as the sampling range expanded, the rate of species accumulation decreased, and the species replacement between pairs of plots slowed down [21]. In the three analyzed scales, the contribution of the species replacement component to the beta diversity of tree species in the suburban secondary evergreen broadleaved forest of Wuchaoshan remained dominant. Environmental heterogeneity and dispersal limitation are two main factors contributing to the scale dependency of the beta diversity [9,10]. They interact to different extents, resulting in varying degrees of differences in species composition among forest communities at different scales, thus influencing the trends in beta diversity [24,32,33]. Similar results were observed by Guo et al. [25] in their study on the spatial distribution patterns and diversity changes of plant species in different vegetation types, topographic features, and altitudes in seasonal rainforests in southern China. They decomposed the species composition differences into two components, species turnover and species richness differences, and found significant variations in the beta diversity among different vegetation types, topographic features, and altitudes in the seasonal rainforests of southern China [25,32]. Both species turnover and species richness differences played essential roles in shaping the beta diversity, and the impacts of environmental factors on species composition varied at different scales [9,10]. Therefore, studying the partitioning of beta diversity in subtropical secondary evergreen broadleaved forests contributes to revealing the spatial distribution patterns and diversity changes of species composition. It provides essential scientific evidence for further understanding the region’s maintenance and conservation of biodiversity [33].

### 3.2. Community Construction Mechanism

Across the three scales investigated, the spatial structure of the environment and pure spatial structure played a dominant role in explaining the proportion of beta diversity in the Wuchaoshan area. This indicates that variables related to the spatial structure are the main drivers shaping the beta diversity patterns in the study area at the investigated scales, likely influenced by diffusion-limitation-driven neutral processes [19,34]. The spatial structure of the environment holds a key position in explaining the variables, because certain environmental factors in the subtropical secondary evergreen broadleaved forest study area, such as soil nutrients and moisture, exhibit spatial heterogeneity [33]. As the analysis scale increases, the explanatory power of purely environmental variables, which have a minor effect on beta diversity at small scales, shows a significant increase. However, the overall explanatory power of variables related to the environment remains lower than that of variables related to spatial structure [25], consistent with the findings of this study. The explanatory power of purely environmental influences increases from only 0.93% at the 10 m × 10 m scale to 11.47% at the 50 m × 50 m scale, representing an increase of over tenfold. This change is believed to be associated with larger-scale environmental filtering-driven niche processes [31]. Habitat filtering and elevation are key factors driving the distribution patterns of beta diversity in the Wuchaoshan area, and in mountainous ecosystems, elevation is considered a decisive factor influencing species distribution.

Across all scales in this study, many of the driving forces behind the beta diversity patterns remain unexplained. In addition to the commonly used variables in research, many other factors can influence the formation of diversity patterns, such as environmental factors indirectly affecting community beta diversity through their impact on microorganisms. Research has demonstrated the significant role of mycorrhizal fungi in maintaining diversity in subtropical forest communities [35]. Zhong et al. found that environmental factors influenced the beta diversity patterns’ formation by affecting tree species rich in arbuscular mycorrhizal fungi. Human activities also extensively impact species composition and beta diversity in suburban forest communities. Activities such as land use changes, logging, and habitat destruction can decrease the beta diversity. In many regions of the southern hilly zone in China, the formation of beta diversity patterns in forest communities is influenced by the long-term interactions between natural factors and human disturbances [21,22]. Therefore, further understanding the formation of beta diversity patterns and their drivers in Wuchaoshan’s secondary evergreen broadleaved tree community is necessary to additionally analyze the community’s environmental component and the factors causing disturbances. It is also important to categorize the collected environmental variables more precisely to reduce their overlap with spatial variables and distinguish between direct and indirect interactions [33].

## 4. Materials and Methods

### 4.1. Description of the Study Site

The study was conducted in Wuchaoshan National Forest Park, located on the western outskirts of Hangzhou, Zhejiang, at the border of Xihu District and Fuyang District, adjacent to the Xixi National Wetland Park and close to the renowned West Lake Scenic Area. It is approximately 22 km away from the city center of Hangzhou. The park is situated within the residual hills of the Tianmu Mountain Range, with an average elevation of 264 m and the highest peak reaching 504 m. The park experiences a subtropical monsoon climate with abundant sunshine and ample rainfall. The average annual temperature is 16.1 °C, with extreme maximum temperatures reaching 41.6 °C and extreme minimum temperatures dropping to −9.4 °C. The hottest month is July, while the coldest month is January, with an average temperature of 3.6 °C. The annual precipitation is 1992.5 mm between May and September, accounting for 61.2% of the total annual rainfall. The yearly evaporation is 1182.6 mm, and the frost-free period lasts around 240 days. The average relative humidity is 76%, and annual sunshine hours range from 1800 to 2100 h. The soil in the area mainly consists of red soil derived from weathered quartz diorite. This forest underwent natural succession after logging in 1950 and is currently dominated by evergreen broadleaved tree species (Appendix A). The dominant forest community in Wuchaoshan is the evergreen broadleaf forest, which includes tree species such as *Schima superba*, *Cyclobalanopsis glauca*, and *Castanopsis sclerophylla*. The deciduous broadleaf tree species in the evergreen broadleaf forest mainly include *Quercus serrata* and *Diospyros japonica*. It is an essential ecological corridor and provides a valuable habitat for numerous plant and animal species.

### 4.2. Construction and Investigation of the Dynamic Monitoring Sample Plots

From 2019 to 2021, a forest dynamic monitoring plot was established and surveyed in a typical secondary evergreen broadleaf forest community within Wuchaoshan National Forest Park, following the CTFS (Center for Tropical Forest Science, [36]) standard. The monitoring plot covers an area of 6 hectares (approximately 300 m × 200 m) and represents a forest with an estimated age of 45 years. The coordinates of the plot’s origin are 120°00′0.60″ E and 30°11′11.37″ N (Figure 3). In the monitoring plot, all woody plants with a diameter at breast height (DBH) greater than or equal to 1 cm were measured. The species name, DBH, tree height, and relative coordinates (the positional coordinates (x, y) of a tree within the 6-hectare plot) were recorded for each individual. These plots cover various habitats and represent the diversity of the suburban secondary evergreen broadleaf forest community. Detailed information on the species composition, abundance, and environmental factors, such as the soil properties and topography were recorded for each plot (Appendix A Table A1 and Table A2).

### 4.3. Data Preparation and Processing

Based on the abundance data of 93 woody plant species in the Wuchaoshan plot at three spatial scales (10 m × 10 m, 20 m × 20 m, and 50 m × 50 m), species abundance plot matrices were constructed with dimensions of 93 × 600, 93 × 150, and 93 × 34 (the number of plot IDs for 93 species at the 10 × 10, 20 × 20 and 50 × 50 scales), respectively (Figure A1). The binary data of the species abundance plot were then transformed using the Hellinger transformation to obtain Euclidean distance properties [31,32].

The environmental variables included terrain and soil factors. The terrain factors included the elevation, slope, convexity, and aspect of each plot at each spatial scale. The elevation was calculated as the mean elevation of the four vertices of each plot. The slope was the average angle between the plane formed by any three angles of the plot and the horizontal plane. The convexity was the difference between the elevation of the plot and the average elevation of all adjacent plots (typically eight neighboring plots, five on the boundary, and three on the corners). The aspect was the mean angle between the plane formed by any three angles of the plot and the actual north direction. Additionally, since the original aspect values ranged from 0 to 2π as a circular variable, they were separately transformed into sine and cosine values for further analysis.

Soil samples were collected in July 2020. The upper litter layer was removed in each 20×20 m surveyed plot, and five randomly located soil samples (0–20 cm depth) were collected using a soil auger. A “quadrant method” retained approximately 1000 g of well-mixed soil samples for soil physical and chemical element analysis. The processed soil samples were sent to the Quantitative Analysis Laboratory at Hainan University for analysis of the soil chemical elements (soil pH, soil organic matter (SOM), total nitrogen (TN), total phosphorus (TP), alkaline nitrogen (AN), available phosphorus (AP), available potassium (AK), bulk density, and moisture content). The values of the soil physicochemical properties in each plot at different study scales were interpolated using ordinary kriging based on semivariograms. A spherical model and weighted least squares fitting were used to obtain the best semivariogram curve, which provided the predicted values of each soil factor within each plot [35]. The calculations and interpolation were performed using ArcMap 10.6.

### 4.4. Analysis of Beta Diversity Patterns and Partitioning Beta Diversity

We employed various statistical methods to assess the beta-diversity patterns in the suburban secondary evergreen broadleaf forest community. First, we calculated the dissimilarity indices based on species abundance data, such as Bray–Curtis indices, to quantify the dissimilarity of species composition among different plots. Then, the beta diversity (BDtotal) of woody plant communities was decomposed into replacement (Repl, the replacement components of beta diversity may be shaped by environmental, spatial, or historical constraints, leading to variations in community composition within forest ecosystems.) and abundance difference (AbDiff, abundance differences are primarily determined by species’ net loss or gain between different locations. When presented along a specific gradient in an ordered manner, smaller communities form subsets of larger communities, often resulting in a nested pattern.) components using the Podani index (The Podani index refers to an index proposed by Podani and colleagues, used to measure dissimilarity or diversity between plots or sites, aiding in the analysis of diversity patterns among ecological communities or sampling sites.) based on the percentage dissimilarity [23]. The calculation formulas [31,32,35] are as follows:BDtotal = (B + C)/(2A + B + C),(1)
Repl = 2 × min(B,C)/(2A + B + C),(2)
AbDiff = |B − C|/(2A + B + C).(3)

In the equation, A represents the total abundance of the shared species between pairwise plots, B and C represent the total abundance of unique species in each pairwise plot, and min (B, C) represents the sum of the minimum quantity among the individual species in each pairwise plot. The measurement and decomposition of beta diversity were conducted using the ‘adespatial’ package in R.

This study employed the Mantel test to address whether the comprehensive environment or spatial position of plant communities significantly influenced the beta diversity of plant communities. Taking the complete environment of plant communities as an example, the Mantel test analyzed whether there was a significant correlation between the matrix of comprehensive environmental conditions and Chao’s distances among the plant communities. Following the completion of the Mantel test, this research also employed partial Mantel tests to explore the impacts of environmental factors and spatial distances on plant community beta diversity. The partial Mantel test builds upon the Mantel test by using a similar concept to partial correlation analysis, controlling for the influence of community spatial distances on plant communities. It examines whether the community’s environmental factors significantly impact the distance matrix of plant communities.

### 4.5. Investigation of Driving Forces

We analyzed the relationships between the beta diversity and environmental variables to investigate the driving forces behind beta diversity in the suburban forest community. We examined the relationship between beta diversity and sampling scales using regression analysis. Additionally, we conducted spatial research to explore the spatial patterns of beta diversity within the study area.

Variance Partition Analysis (VPA) was conducted at three spatial scales to investigate the contributions of different explanatory variables (significant environmental and spatial structure variables) to the beta diversity in the Wuchaoshan community. Before conducting the variance partition analysis, the environmental and spatial structure variables were screened to eliminate redundancy and obtain an economic model. This step was performed using distance-based Redundancy Analysis (dbRDA).

VPA decomposes the total variation of beta diversity and its components into four parts for explanation: (a) pure environmental effects, (b) environmental spatial structure effects, (c) pure spatial structure effects, and (d) unexplained variation. The (a + b) part is related to the ecological niche processes dominated by environmental filtering, while the (c) function is associated with the neutral processes dominated by dispersal limitation. Some studies have also indicated that the (c) part can be influenced by spatially structured environmental variables and the spatial structure effects of community dynamics that are not yet clearly understood [25,26,32,34].

The dbRDA analysis, VPA analysis, and significant explanatory variable selection before VPA were performed using the ‘vegan’ package in R. The dbrda function was used for dbRDA, the ordiR2step function was used for significant explanatory variable selection, and the carport function was used for VPA.

### 4.6. Statistical Analysis

All statistical analyses were performed using appropriate software packages, including R and statistical analysis systems. We used standard statistical techniques to test the significance of the results and evaluate the strength of the relationships observed.

## 5. Conclusions

This study found that the beta diversity of the secondary evergreen broadleaved forest community in Wuchaoshan exhibits a decreasing trend as the spatial scale increases. Within the analyzed scales of this study, species turnover consistently constitutes the dominant component in forming beta diversity patterns, and its contribution increases with scale. On the other hand, the proportion of the species abundance difference component decreases with scale. Both the dispersal limitation and environmental filtering contribute to the formation of beta diversity patterns in the secondary evergreen broadleaved forest community of Wuchaoshan, but their roles vary at different scales. At smaller scales, dispersal limitation plays a more prominent position, while environmental filtering becomes increasingly influential as the scale expands and the impact of dispersal limitation diminishes. This study on the beta diversity of woody tree species in the secondary evergreen broadleaved forest of Wuchaoshan partially supplements and enhances species coexistence theory in related forest communities. It provides valuable insights into the scientific management and operation of forest communities, primarily suburban and urban forests in the region. It is important to acknowledge certain limitations of this study. The results from a specific geographic region may not directly apply to other areas with different ecological characteristics. Additionally, the study focused on a particular type of forest community, and the findings may not be generalizable to other types of ecosystems. Future research should consider expanding the study area and incorporating a broader range of environmental factors to understand the beta diversity dynamics in suburban forests better.

## Figures and Tables

**Figure 1 plants-12-03057-f001:**
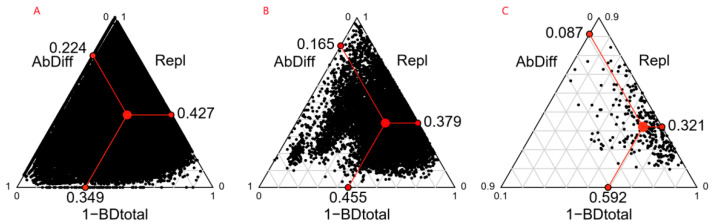
Triangular plots of beta diversity and its two components at different spatial scales. Note: from left to right, the scales are 10 m × 10 m (**A**), 20 m × 20 m (**B**), and 50 m × 50 m (**C**); each black dot represents a plot pair; their positions are determined by BDtotal, Relp, and AbDiff values, with the sum of the three values equaling 1; the red dot inside the ternary plot represents the centroid of all the black dots, and the red dots on the boundary lines represent the average values of the three components.

**Figure 2 plants-12-03057-f002:**
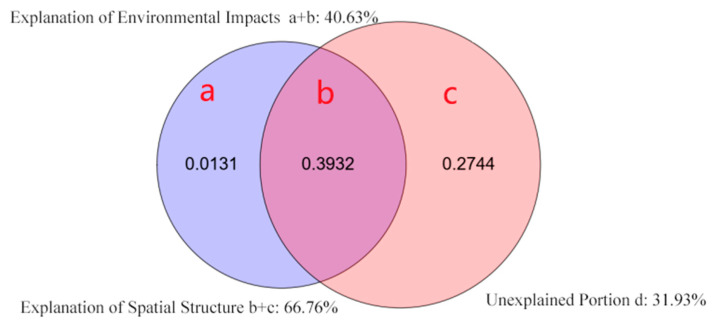
VPA results of the BD total between two sets of explanatory variables: environment variables and dbMEM eigenfunctions for 20 × 20 m scale in the sample plot. Note: (a) pure environmental effects, (b) environmental spatial structure effects, (c) pure spatial structure effects, and (d) unexplained variation. The (a + b) part is related to the ecological niche processes dominated by environmental filtering, while the (c) function is associated with the neutral processes dominated by dispersal limitation.

**Figure 3 plants-12-03057-f003:**
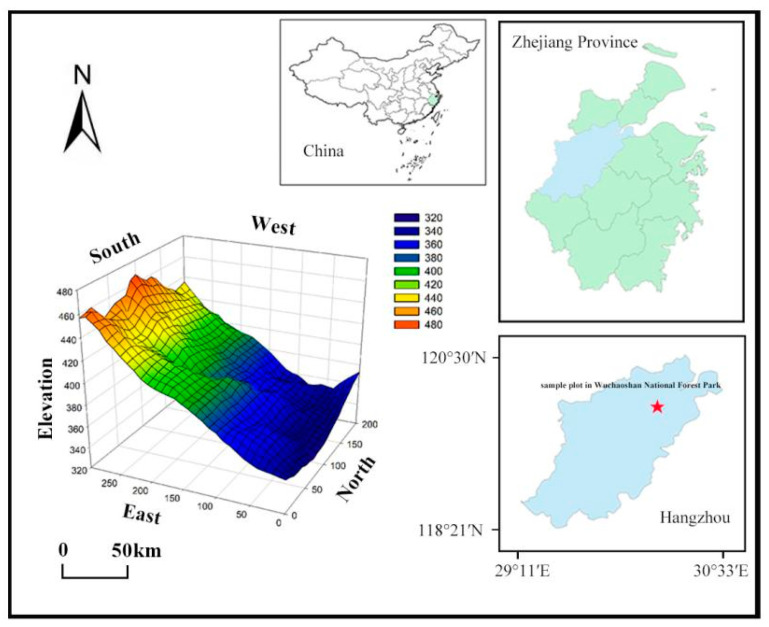
Location and 3D topographic map of the forest dynamic monitoring plot in Wuchaoshan National Forest Park. Note red asterisk is the sample.

**Table 1 plants-12-03057-t001:** Variation partitioning results for different spatial scales in the sample plot.

Scale	Number of Samples	(a)	(b)	(c)	(a + b)	(b + c)	(d)
10 m × 10 m	600	0.0093	0.2796	0.3890	0.2889	0.6686	0.3221
20 m × 20 m	150	0.0131	0.3932	0.2744	0.4063	0.6676	0.3193
50 m × 50 m	24	0.1147	0.5011	0.1723	0.6158	0.6734	0.2119

## Data Availability

Not applicable.

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
