# Peer review of "Environmental Filtering and Dispersal Limitations Driving the Beta Diversity Patterns at Different Scales of Secondary Evergreen Broadleaved Forests in the Suburbs of Hangzhou"

_plants, 2023, doi:10.3390/plants12173057_

Round 1

Reviewer 1 Report

Abstract should contain information about the methods used in regard to plots, what was measured etc.

Title and elsewhere – the forest naming “sub-tropical sub-urban secondary evergreen broad-leaved forest” is really long and not clear. I understand authors want to provide an informative name for the forest under study but the chosen one is too complicated.

Title – It seems to me that this manuscript is an evaluation of the effect of sampling scale on beta diversity and environmental factors. It is a limited study in terms of data, size and geographically limited, therefore, it is an addition to this topic and not a definitive conclusion. As such, title, abstract and conclusions should reflect it.

Methods should be placed before Results.

There are many aspects to be fixed in Methods, including full description of the types of analysis and the inclusion of a significance testing.

L8-9 – following the previous comment, here it seems a bit clearer that the point is being a periurban forest more than being subtropical, broad-leaved or even secondary.

L12 – driving forces in relation to what?

L34-35 – What is the novelty used in the mentioned citation in relation to “large” scale analysis? It has been consolidated plots of 1 ha in tropical forests. How did the mentioned study innovate on that? As it is, the statement is too vague.

Section 2.1 (L121-130) – It must be rewritten. There are values cited that make no sense or at least one cannot easily (it at all) find them or understand them.

Figure 1 – letters should be described in captions and placed in the figure itself.

L138-139- Authors should clearly justify the use of b+c to explain factors influence. I have profound doubts about the use of b+c as authors did to interpret the results.

It represents the shared fraction of variation explained when the two are included in the model, meaning it is the portion of variation that cannot be attributed to a or b separately. In other words, the variation partitioning cannot be disentangled.

As a result, authors are somehow doubling the effect of b by adding c. Remember – c is the shared fraction of variation explained when the two (a, b) are included in the model.

L237 – Include map with province, park, and plots.

L252 – Please italicize scientific names.

L260 – What is the CTFS (Center for Tropical Forest Science) standard? Citation, explanation.

L265 – “relative” coordinates to what? Or the coordinates were simply obtained with a GPS.

L267-269 – Include in the appendix the main characteristics that were mentioned in text for the plot.

L147-234 - Whole section should be rewritten considering the significance testing. Also, there are some clear overstatements that do not seem to be based on the results (apart from the problem with the adding b+c).

L147-234 - Paragraphs are way too long.

L271-275 – Authors must explain why the values were chosen (93×600, 93×150, and 93×34m), and how they could be observed in the field. Explain in detail and add a picture of the different sub-plots in relation to the main plot.

L275 – Why did authors use Hellinger transformation. Cite reference for that application. As it is not a widespread used transformation, it should be explained so readers can follow the rationale well informed.

L277-278 and after – “Elevation {and other factors} was calculated as the mean elevation of the four vertices of each plot.” Which plot size are authors referring to? Whatever the size is, it will affect the overall sub-plot analysis, i.e., correlation between environmental factors x plot x diversity. Fully explain thresholds and their impacts on analysis/results.

L286 – “In each surveyed plot” – Again: Which plot are authors referring to?

L288 – What is a “quadrant method”? How deep were the soil samples? Mixing different depths certainly affects the results – how did authors consider this?

L286-297 – Averaging soil results for large plots (still not clear how big the plots are) will certainly affect the analysis. First, explain the size of plots, secondly, explain how averaging soil (and all other factors) for a plot affects analysis and results.

L304-307 – Although some of the most important references are cited, authors should explain the Podani index as it is seldom seen in publications about diversity.

L305- replacement (Repl) and abundance difference rationale must be explained here. Again, readers should be able to understand this method as it is not commonly used.

Section 4.4 – Authors should include testing for significance!

English is in general good. Results have paragraphs that are way to long and should be splitted for easy of understanding.

Reviewer 2 Report

This article presents an analysis of the beta-diversity of subtropical suburban secondary evergreen broad-leaved forest communities. Overall, the article is technically sound; however, there are a few queries and remarks that need explanation.

The Results section is just two paragraphs. This section should be significantly enhanced, and study findings should be added.

Why did the authors not include a map showing the locations of the study sites? This information will significantly enhance how the material is perceived.

What forests (first of all, species composition) grew in these territories previously? What caused the forest communities to change? Did fires or logging force the communities to change?

How did secondary woods get their start? Did they grow because of their own renewal (caused by the lingering seed bank in the soil)? Or were they cultivated artificially?

What now stands in for these forests? 93 tree species (line 273) were discovered during this study, or was this information already known to other researchers? Which woody plant species are represented by the first (main) layer of the tree stand? What is the average height and diameter of the first layer of the tree stand? Are these closed forests or not? Which woody plant species were confined to the undergrowth?

Why do the authors not include information on the qualitative and quantitative traits of the woody species that have been found and researched? Why aren't the average height and diameter for each of the discovered species represented? Why is the average amount of woody plant specimens that have been found for each species (pieces per square metre) not reported? The "Results" section or an article annexe should contain all of this information.

The findings of soil research (physico-chemical properties) should be presented by the authors.

Figure 2. All inscriptions in the figure must, first and foremost, be in English. Second, what is the significance of this figure in the context of the article? The picture just shows the slope profile; it says nothing about how the woody plant species are spatially distributed within the study area. The figure needs to be supported by data, or it can be deleted without degrading the quality of the information presented in the article.

“…were constructed with dimensions of 93×600, 93×150, and 93×34, respectively…” (Lines 273). Table 1 shows 24 in row 3 (number of samples). Which of these data are correct?

Round 2

Reviewer 1 Report

I suggest another checking for minor mistakes.

For example,

1) Appendix 1 mentions "abundant" and "important values" but i assume authors are referring to "abundance" and "importance value".

2) Importance value is originated from the calculation of density, frequency and dominance. Neither of the three indexes were mentioned, their values presented (even less the data that allow for such calculation)

3)L295-297 - in the revised version of the manuscript reads "Detailed information on species composition, abundance, and environmental factors such as soil properties and topography were recorded for each plot (Appendix 1).

But notice that Appendix 1 does not show data/results for soils, topography or any environmental factor.

4) I still think that figure 2 is confusing. What does it mean a, b and c? Please add information to the captions.

Minor spellings should be checked.

Reviewer 2 Report

The authors answered all the questions and comments. The manuscript has been carefully revised. I believe that the manuscript can be accepted for publication.
